

# Assessing Atmospheric Gravity Wave Spectra in the Presence of Observational Gaps

Mohamed Mossad [1], Irina Strelnikova [1], Robin Wing [1], and Gerd Baumgarten [1]

[1]Leibniz Institute for atmospheric physics, Kühlungsborn, DE

**Correspondence:** Mohamed Mossad (mossad@iap-kborn.de)

**Abstract.** We present a thorough investigation into the accuracy and reliability of gravity wave (GW) spectral estimation methods when dealing with observational gaps. GWs have a significant impact on atmospheric dynamics, exerting influence over weather and climate patterns. However, empirical atmospheric measurements often suffer from data gaps caused by various factors, leading to biased estimations of the spectral power-law exponent ($\beta$). This exponent describes how the energy of GWs changes with frequency over a defined range of GW scales. In this study, we meticulously evaluate three commonly employed estimation methods: the Fast Fourier Transform (FFT), Generalised Lomb-Scargle periodogram (GLS), and Haar Structure Function (HSF). We assess their performance using time series of synthetic observational data with varying levels of complexity, ranging from a single sinusoid to superposed sinusoids with randomly distributed wave parameters. By providing a comprehensive analysis of the advantages and limitations of these methods, our aim is to provide a valuable roadmap for selecting the most suitable approach for accurate estimations of $\beta$ from sparse observational datasets.

**Keywords.** spectra, gravity waves, lidar, Lomb Scargle, Fourier, Haar

## 1 Introduction

Gravity waves (GWs) are ubiquitous phenomena that play a crucial role in the dynamics of the Earth's atmosphere, where they impact weather and climate patterns (Hines, 1960; Ern et al., 2018). Various sources, including convection, topography, and jet streams generate these waves (Crowley and Williams, 1987; Fritts, 1989). As they propagate through the atmosphere, they can break and mix with the surrounding atmosphere, redistributing their energy and momentum. This leads to significant changes in the atmospheric thermodynamics and large-scale circulation patterns of the atmosphere, including wind speeds and temperature gradients (Lindzen, 1981; Holton, 1983; Fritts and Alexander, 2003). Observations of these meteorological variables reveal that GWs exist for the most part in the form of a spectrum of superposed waves within a wave packet, and occasionally as quasi-monochromatic waves (Maekawa et al., 1984; Eckermann and Hocking, 1989). To understand the physical processes that govern the generation, propagation, and dissipation of these wave packets, it is often useful to examine their spectral prop-





erties such as their frequencies, amplitudes, and scales (Axford, 1971; Fritts and VanZandt, 1993).

On that note, Vanzandt first introduced the concept of a 'universal atmospheric GW spectrum' (VanZandt, 1982). This spectrum facilitated efficient parameterizations of how GWs affect the mean atmospheric state (Babu et al., 2008). For instance,

the spectra of GWs are often used in model parameterization, including source spectra parameterization, Lagrangian spectral parameterization, and subgrid-scale parameterization, enabling the simulation of the dynamics of the middle and upper atmospheres (Beres et al., 2005; Song and Chun, 2008; Houchi et al., 2010). Overall, accurate predictions of GW activity can improve weather forecasting, while they contribute significantly to climate modelling in parameterizing physical processes like turbulence and mixing (Alexander et al., 2002; Smith, 2012; Liu et al., 2014).

This universal GW spectrum exhibits a power law scaled by an exponent (or slope) $\beta$, which describes the rate at which wave energy changes with its wave number (or frequency). The basis for this spectrum for atmospheric GWs is not only supported by a strong foundation in theoretical works (Dewan and Good, 1986; Weinstock, 1990; Hines, 1991; Dewan, 1994; Gardner, 1994), but also in observational studies (Smith et al., 1987; Fritts et al., 1988; Gardner et al., 1995; Nastrom et al., 1997; Zhang et al., 2006, 2017). These values of $\beta$ do not only depend on the type of spectra (e.g. temporal, horizontal, or vertical

wavenumber) but also the geophysical variables measured (e.g. temperature, horizontal or vertical wind etc.), see Tab. A1 for a summary. Thus, an accurate estimation of $\beta$ is essential to validate different theoretical predictions of GW power spectral densities (PSD) (Dewan and Grossbard, 2000), and improve climate models and weather forecasts (Lindgren et al., 2020).

Determining $\beta$ from empirical atmospheric measurements is challenging due to various factors, such as the inevitable presence of data gaps, observational noise, and the finiteness of data length and resolution. Data gaps can occur for numerous reasons,

including: instrumental errors, data transmission issues (e.g. due to weather conditions like clouds in the case of lidar), and signal interference (in the case of radar). When gaps exist in multiscale time series, data points representing certain frequencies are lost, which distorts the spectra and introduces significant bias in the estimation of $\beta$ (Brown and Christensen-Dalsgaard, 1990; Rigling, 2012). To minimise the effect of these gaps on the spectra, data-filling schemes are often applied. Though linear interpolation is usually used to fill in these gaps (Meisel, 1978; Lepot et al., 2017), even adaptively implemented interpolators

produce artefacts into the time series at low gap percentages (GPs), which contribute additional bias in the spectra (Schulz and Stattegger, 1997; Hall and Aso, 1999). Bias in spectral estimates can also be caused by other relevant sources, such as spectral leakage, steep spectra ($\beta > 2$), and in-signal components with larger periods than the observed time span $T$ (Klis, 1994).

In this paper, we systematically quantify the advantages and limitations of estimation methods of GW spectra in handling these error sources. We also propose a procedure for selecting unambiguously suitable methods for $\beta$ estimation. Two com-

monly used methods are considered, namely the Fast Fourier Transform (FFT) (Cooley and Tukey, 1965) and the Generalised Lomb-Scargle periodogram (GLS) (Zechmeister and Kürster, 2009), as well as the fairly recent Haar Structure Function (HSF) (Lovejoy and Schertzer, 2012). The FFT is the standard method to analyse spectra of evenly sampled data, while the GLS and HSF are specifically known to handle unevenly sampled data. In an effort to closely mimic real observations of GWs, we simulate time series data with varied levels of complexity, beginning with signals consisting of single sinusoids and increasing

in complexity to a superposition of sinusoids with randomly distributed scales (and frequencies).

Previous studies have investigated these spectral methods and others for estimating power-law spectra and compared their per-





formance on synthetic and observed data. For instance, Zhan et al. found that using FFT of linearly interpolated signals is the best approach to analyse radar wind data at $50\%$ GP (only for the case of $\beta = 5/3$) compared to the correlogram and Lomb-Scargle (LS) (Zhan et al., 1996). However, a quantitative analysis of the effect of changing $\beta$ or the GP was not conducted.

Similarly, Munteanu et al. showed that FFT outperforms LS, Z-Transform, and Discrete Fourier Transform in estimating $\beta$ from Venus' magnetic field data (Munteanu et al., 2016). Although, the effect of changing $\beta$ was not considered either since the power-law spectra were not simulated. In contrast, Hébert et al. found that the HSF consistently surpassed other methods in estimating $\beta$, without the need to interpolate the gapped (simulated paleoclimate) data for $\beta \in (0,3)$, except the case of $\beta \in (-1,0)$, where they concluded that LS would be the best option (Hébert et al., 2021). Nonetheless, the impact of altering

the GP was not quantitatively presented, instead, the skewness of the gaps (Gamma) distribution was used as a parameter to refer to the irregularity of the time series.

The rest of the paper is organised as follows: in Sect. 2, we describe the methods used in our study, including a description of FFT, GLS, and HSF. In Sect. 3 we introduce the data simulation procedures. In Sect. 4 we discuss data processing. In Sect. 5 we present the results of our simulations, comparing the performance of these methods in different scenarios. In Sect. 6 we

discuss the implications of our findings, and provide recommendations for spectral analysis of GW time series with data gaps. Finally, in Sect. 7, we present a summary of our relevant results and conclusions.

## 2 Spectral Methods

### 2.1 The Fast Fourier Transform

The FFT is the most commonly used method for estimating frequency spectra of evenly sampled data (Cooley and Tukey,

1965). It enables the approximation of a time series sampled from a continuous distribution over discrete time steps, through a series of complex sine and cosine waves with varying frequencies. Under the assumption of a unit sample interval, the (forward) FFT transforms a time series $z_n$ of length $N$ from its original domain (time or space) into a set of coefficients $Z_k$ in the (temporal or spatial) frequency domain by employing the relation:

$$Z_k = \sum_{n=0}^{N-1} z_n e^{-2\pi i k n/N}, \quad k = 0,1,...N-1. \tag{1}$$

In our work, the FFT will serve as the benchmark spectral estimation method. The expected Fourier transform of a discretized signal is given by the convolution of the true transform and the transform of a Dirac comb window function designating those measurement times (Vanderplas, 2018). In the case of gapped data, the symmetry in the Dirac comb is destroyed, causing the resulting transform to be noisy with incorrect peak positions and heights. Consequently, the true transform of gapped data will not be recoverable. This disadvantage can be bypassed by applying data reconstruction methods such as interpolation, sparse

approximation, etc., to approximate the true Fourier transform (Babu and Stoica, 2010). Unfortunately, these reconstruction



methods can introduce artefacts to the signal, which depend on the distribution of the gaps and their sizes (Munteanu et al., 2016).

## 2.2 The Generalised Lomb-Scargle Periodogram

The GLS periodogram developed by Zechmeister and Kuerster offers a method for estimation of the PSD of unevenly sampled
time series (Zechmeister and Kürster, 2009). It is a generalisation of Lomb's least-squares approach (Lomb, 1976) which is equivalent to the modified Schuster's periodogram (Schuster, 1898; Scargle, 1982) (based on the FFT) in the case of evenly sampled data. The GLS produces a spectrum by least-squares fitting a model of a weighted sinusoid given by

$$y(t) = a\cos\omega t + b\sin\omega t + c \tag{2}$$

to the time series at each sampled frequency $\omega$. The offset $c$ compensates for the assumption that the mean of the time series $\overline{z}$
is equal to the mean of the fit $\overline{y}$. This floating-mean approach is advantageous, considering that the mean of a periodic signal may change statistically, especially for small $N$ (Ferraz-Mello, 1981). Furthermore, the purpose of using weighted sums is to account for the observational noise for which the original LS does not.

The LS method has often been used to seek dominant periodic frequencies or cycles (Zhang et al., 1993; Pichon et al., 2015; Rao et al., 2017), analyse seasonal changes of significant modulations of GW fields (Beldon and Mitchell, 2010), and estimate
the spectral indices $\beta$ and amplitudes of GW power-law spectra (Hall and Aso, 1999; Zhang et al., 2006; Guharay and Sekar, 2011; Qing et al., 2014). In addition, LS is known as the most efficient method for estimating the variance in both gapped and non-gapped stationary time series with a single-sinusoidal periodicity, without the need to fill in missing data (Marinna et al., 2019). In contrast, Vio et al. found that the LS is neither reliable for analysing semi-periodic nor aperiodic signals with non-stationary noise or signals made of more than one sinusoid, without additional steps (Vio et al., 2010).


## 2.3 The Haar Structure Function

The HSF is a mathematical tool used in conducting scaling analysis of signals (Lovejoy and Schertzer, 2012), which is based on the Haar wavelet (Haar, 1910). It is a simple yet powerful method for decomposing a signal $x(t)$ whose power spectral density exhibits a power law, i.e. PSD $\propto \tau^H$ with a scaling (Hurst) exponent $H$, over a scale (lag) $\tau = 1/f$, into fluctuations
$\Delta x = x(t+\tau) - x(t)$. The first-order Haar fluctuations $\mathcal{H}_\tau$ at a lag $\tau$ are defined by the relation:

$$\mathcal{H}_\tau(x(t')) = \frac{2}{\tau}\left|\sum_{t+\frac{\tau}{2}<t'<t+\tau} x(t') - \sum_{t<t'<t+\frac{\tau}{2}} x(t')\right| \tag{3}$$



The q-th order structure function $S_q(\tau)$ is then obtained as an approximation by ensemble averaging these fluctuations as follows:

$$S_{\mathcal{H},q=1}(\tau) = \langle \mathcal{H}_\tau(x(t)) \rangle \approx \tau^{qH-K(q)} \tag{4}$$

In the quasi-Gaussian case, the moment scaling function is $K(q) \approx 0$, so for the first-order structure function ($q = 1$), only $H$ determines the scaling of mean fluctuations.

A power spectrum which follows a power law PSD $\propto \tau^\beta$ is related to the Hurst exponent by $\beta = 1 + 2H - K(2)$, here $q = 2$ because the power spectral density is a second-order moment. Thus, under the quasi-Gaussian approximation, we re-scaled the HSF to a comparable scale to the PSD using the relation:

$$S_{\mathcal{H},q=1} \propto \tau^H \approx \tau^{\frac{\beta-1}{2}} \tag{5a}$$

$$S_{\mathcal{H},q=1} \cdot \tau^{1/2} \propto \tau^{\beta/2} \tag{5b}$$

$$S_{\mathcal{H},q=1}^2 \cdot \tau \propto \tau^\beta \propto \text{PSD} \tag{5c}$$

The HSF is particularly suitable for estimating the scaling exponent of time series with $H \in (-1, 1)$ or $\beta \in (-1, 3)$. This range of $\beta$ values covers the vast majority of atmospheric processes from weather (where $\tau < 10$ days, and $1 < \beta < 3$) to macroweather systems (where $10$ days $< \tau < 10 - 30$ yr, and $-1 < \beta < 1$). The HSF also possesses the advantage of handling unevenly sampled data, which is a consequence of the fact that it is computed by taking the mean of absolute fluctuations. Nevertheless, the HSF is not employed to estimate the amplitude or the frequency of sinusoids, since it only measures how much frequency components contribute to the total variance. The Python code implementation of the HSF is readily accessible (Mossad, 2023), which was derived from the R code originally developed by Raphaël Hébert (Hébert, 2021).

## 3  Data Simulation

In this section, we present the simulation procedures used to generate time series similar to actual GWs measurements. In measurements, GWs can exhibit various behaviours, ranging from superposed waves within wave packets with multiple frequencies, amplitudes, and phases to more coherent quasi-monochromatic waves (Maekawa et al., 1984; Eckermann and Hocking, 1989; Sica and Russell, 1999). In Sect. 3.1, we simulate single sinusoids to mimic quasi-monochromatic waves, in which the goal is to accurately estimate the correct frequency and amplitude of each signal. In Sect. 3.2, however, we simulate time series composed of a superposition of scales and amplitudes. This simulation allows us to produce spectra which follow power laws, whose exponents $\beta$ are used to assess the bias of the spectral analysis methods.

By analysing both simulations, we can determine the accuracy of each of the methods at different levels of signal complexity, and identify potential limitations and sources of error in the analysis of GWs spectra. Random gaps are then introduced to re-





semble observational gaps for both simulations. The units and values of the variables used in this simulation have been chosen to represent average values or ranges characteristic of typical GW time series.

### 3.1 Single-Sinusoid Simulation

Quasi-monochromatic GWs can be observed under specific conditions where a single frequency dominates other components (Muraoka et al., 1988; Swenson et al., 1999). This kind of GWs can be approximated as an evenly sampled single sinusoid $x(t) = A \sin(2\pi f t + \varphi)$ with a known frequency $f$, phase shift $\varphi$, and amplitude $A$, at time $t$. For both simulations, the time resolution $\Delta t$ is $5\,\mathrm{min}$ with a total span of $T = 6\,\mathrm{h}$, as this resolution and duration align with the average values of lidar measurements commonly used in atmospheric studies (Gardner et al., 1995; Gerding et al., 2008). As a result, the number of points $N$ in each signal is equal to $\frac{T}{\Delta t} = 72$. Each simulated sinusoid has an arbitrary amplitude of $A = 4\,\mathrm{K}$ and a randomly chosen frequency $f$ from the set $1/\{6, 3, 1.5, 1, 0.5, 1/3\}[\,\mathrm{h}^{-1}]$. Changing the frequency serves as a test to examine whether the bias of the methods is frequency-dependent. The phase shift $\varphi$ is also randomly chosen but from a uniform distribution within the interval $[0, 2\pi]$.

The amplitude of the time series is equivalent to the estimated height of the main peak in the spectrum. It is computed from the FFT coefficients (Eq. 1) as $A_{\mathrm{FFT}} = \max_k \left( \frac{2|Z_k|}{N} \right)$ and from the GLS fit coefficients (Eq. 2) as $A_{\mathrm{GLS}} = \max_k \left( \sqrt{a_k^2 + b_k^2} \right)$. The frequency of that peak $f_k$ corresponds to the estimated frequency of the signal. As a metric for the accuracy of estimation of the true values of frequencies and amplitudes, we used the relative bias given by:

$$\text{Relative bias} = \frac{\text{value estimated} - \text{value expected}}{\text{value expected}} \quad . \tag{6}$$

Since real data is susceptible to observational noise, it is crucial to consider the case where white noise is added to the simulated signal as a random variable $r(t)$ from a standard normal distribution. Here, the signal-to-noise ratio is defined by $\mathrm{SNR} = A^2/2\sigma_r^2$, where $\sigma_r^2$ is the noise variance (Horne and Baliunas, 1986). To strike an appropriate balance between capturing meaningful noise characteristics and minimising low SNR bias, an average SNR value of $8$ is chosen for this simulation.

### 3.2 Spectral Power Law Simulation

As reported before (see Table A1), the universal spectra of GWs are characterised by a power law, i.e. $\mathrm{PSD} \propto 1/f^\beta$. On that account, we are interested in estimating the value of $\beta$ of simulated time series whose spectra would have $\beta \in \{-1, 0, 1, 5/3, 2, 2.5, 3\}$ and comparing it with the true value. The simulated evenly sampled time series $x(t)$ consists of a sum of $M$ sinusoids, each with frequency $f_i$ and power law amplitudes, $A_i = f_i^{-\beta}$ as follows (Rice, 1944)

$$x(t) \sim \sum_i^M \sqrt{A_i} \sin(2\pi f_i t + \varphi_i) = \sum_i^M (f_i)^{-\beta/2} \sin(2\pi (f_i) t + \varphi_i) \quad . \tag{7}$$



We used for our simulation $M = 35$ for each time series. It is driven by the objective of reconciling the inclusion of a minimum of 20 waves based on observational (Sica and Russell, 1999) and modelling suggestions (Dewan, 1994; Hamilton, 1997), while simultaneously incorporating a sufficiently large number of waves to mitigate random spectral errors (Keisler and Rhyne, 1976). This approach enables the demonstration of the power law spectrum without the need for excessive averaging. The frequencies $f_i$ are statistically independent uniformly distributed random values, selected within the range $\left[\frac{1}{T}, \frac{1}{2\Delta t}\right] = \left[\frac{1}{6\,\text{h}}, \frac{1}{10\,\text{min}}\right]$. Hence, the time series is composed of non-harmonic components and there are no favoured frequencies, which is a better approximation of atmospheric GWs than an idealistic case where frequencies are only integer multiples of a fundamental frequency. Here, $x(t)$ is proportional to the square root of the amplitudes $A_i$ since $\beta$ is estimated from PSD-normalised spectra, which are the squared modulus of the amplitudes.

The PSD is obtained by the FFT using the relation $\frac{2Z_k Z_k^* \Delta t}{N}$, where $Z_k^*$ is the complex conjugate of $Z_k$. This definition is equivalent to the GLS spectrum $\frac{N|a_k^2 + b_k^2|\Delta t}{2}$ of evenly sampled data. The HSF is however normalized according to Eq. 5c to estimate a comparable scale to the PSD. The bias of $\beta$ estimation is defined for this simulation as

$$\beta \text{ bias} = \text{value estimated - value expected} \quad . \tag{8}$$

### 3.3 Gaps Simulation

After creating a time series with the desired spectral properties, gaps are introduced by randomly removing data points (except both endpoints), assuming that all data points are equally probable to be removed (i.e. a uniform distribution). Based on the simulated GP $p$ in the data, an integer number of random points $N_G = N - \frac{N \cdot p}{100}$ is removed. Thus, a $0\%$ GP means that no points were removed, while a $50\%$ GP means that 36 points were randomly removed, since $N = 72$. To assess the dependence of bias in spectral analysis methods on the gaps, we conducted simulation runs spanning GPs ranging from $0\%$ to $90\%$ in increments of $10\%$ for each time series analysed. Each of these simulation runs was repeated 1000 times at each GP increment to ensure the statistical significance of our results since the frequencies, phases, and gaps are randomised. Then we computed the average values of the estimated amplitude and period (for the single-sinusoid simulation), and $\beta$ (for the spectral power law simulation).

### 4 Processing Steps

Before applying spectral methods on the generated time series from the simulations in Sec.3, the following steps are taken:

- The time series is first interpolated using the original time step $5\,\text{min}$, this is only necessary for FFT.

- The mean of the signal $\overline{z}$ is subtracted to account for the zero-frequency component of the Fourier transform $Z_0$.

When computing the spectrum for a time step $\Delta t$, the frequency grids of all methods are defined as follows:

- The frequency range spans from $\frac{1}{T}$ to $\frac{1}{2\Delta t}$.



- The frequency spacing is given by $\Delta f = \frac{1}{T}$.

In the presence of gaps where $\Delta t$ is not constant, the Nyquist frequency is then defined as:

- $f_{Ny} = \frac{1}{2p}$, where $p$ is the largest value that allows $t_i = t_0 + n_i p$ to be possible for all $t_i$, and $n_i$ are integers (Eyer and Bartholdi, 1999). This value corresponds to the same Nyquist frequency of our non-gapped data, which is $0.1\,\mathrm{min^{-1}}$.

- This approach is more appropriate for GLS and HSF than applying a "pseudo-Nyquist" limit based on an average or a minimum value of $\Delta t$ (Scargle, 1982; Vanderplas, 2018).

To estimate $\beta$, the spectra are fitted by taking the following steps:

- A maximum likelihood estimator (MLE) is employed to determine the fit parameter $\beta$.

- The MLE fit involves minimising the negative log-likelihood function $-\ln L(\overrightarrow{O})$ of observations $O_i$ at frequency $f_i$ using the equation:

$$-\ln L(\overrightarrow{O}) = \sum_{i=1}^{n} \ln \langle O_i \rangle + \frac{O_i}{\langle O_i \rangle} \quad , \tag{9}$$

where $\langle O_i \rangle$ refers to the power law model $c(\frac{1}{f^\beta})$ being fitted, with $c$ as a normalisation coefficient (Duvall and Harvey, 1986).

- The MLE fit is recommended over least squares regression because the latter assumes a Gaussian distribution of periodogram residuals, leading to a biased estimate of $\beta$ (Clauset et al., 2009).

## 5 Results

### 5.1 Single-Sinusoid Signal

First, we show an example of the time series generated by the simulation described in Sect.3.1, which consists of a $0.5\,\mathrm{h}$-wave in a 6-hour time series. As can be seen in Fig.1a, in the absence of gaps, an accurate estimation of the $4\,\mathrm{K}$ amplitude and the $0.5\,\mathrm{h}$ period of this signal is acquired by both the GLS and FFT. In addition, the spectra obtained by the FFT and GLS are (as expected) equivalent in the case of evenly sampled data (Scargle, 1982). When the random gaps replaced $50\%$ of the data points, a significant difference between the amplitude spectra is observed, see Fig.1b. Both methods still provide an accurate estimate of the signal's period. Linear interpolation of the $50\%$ gapped signal preserves the structure of the wave but loses some of the high-frequency components, which leads to a significant underestimation of the amplitude by $43\%$ in the FFT spectrum. In contrast, the amplitude of the highest peak in the GLS spectrum is not affected by the gaps and has not changed from the expected value.

When comparing average relative period bias for different simulation periods, Fig.2a shows that GLS demonstrates no period





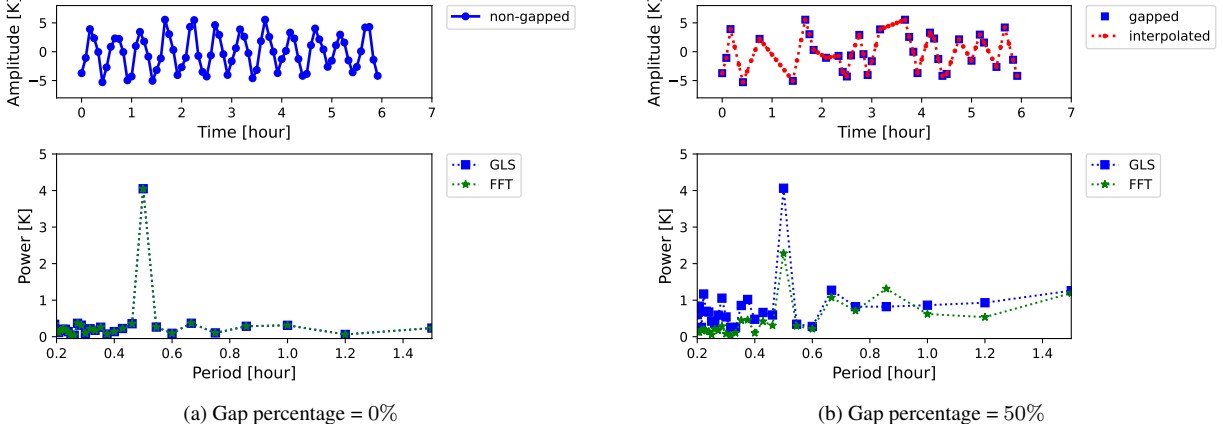

(a) Gap percentage = 0%                                    (b) Gap percentage = 50%

**Figure 1.** Time series of a $0.5\,\mathrm{h}$-wave in a 6-hour observation time generated according to Sect.3.1. (a) shows the time series (upper left) and its temporal amplitude spectrum (lower left) in the absence of gaps. (b) shows the time series (upper right) and its temporal amplitude spectrum (lower right) after the addition of random gaps.

bias below $80\%$ GP and a negligible bias beyond (within $\pm 20\%$ deviation interval). Fig.2b shows, contrarily, that the smaller

the simulated period of the signal, the more FFT overestimates it at GPs larger than $40\%$. This is due to linear interpolation replacing the removed high-frequency components with lower-frequency ones (i.e., aliasing), which eventually dominate as the GP gradually increases. To put these results in perspective, at $70\%$ GP, FFT inaccurately determines the period of a $0.5\,\mathrm{h}$ wave as $2.93\,\mathrm{h}$ (i.e. overestimates it by $486\%$). Given that there are quite few data points left in the time series at $70\%$ GP, it is expected that FFT is not able to recover the correct period, however, GLS is still capable of obtaining the exact value of the

period (i.e., an error of $0\%$). Not only is the GLS a much better estimator of the period on average, but also since its standard deviation remains trivially small until $80\%$ GP, it is a more reliable choice on a case-by-case basis as well.

Similarly, the FFT's amplitude bias experiences clear dependency on the frequency of the signal, while GLS demonstrates a negligible amplitude bias at GP below $80\%$, see Fig.3. In particular, the results show that the mean estimated amplitude from the FFT spectrum deviates exponentially from the expected value as the frequency of the signal increases, even at gap percent-

ages below $50\%$. In addition, the standard deviation of amplitude estimation by GLS increases as the GP increases, although it remains within $\pm 10\%$ deviation interval up to $80\%$ GP. On the contrary, the standard deviation of amplitude estimation by FFT significantly increases as both the frequency of the signal and GP increase, implying that FFT is more inconsistent and highly sensitive to missing data, especially for high-frequency signals.

Overall, GLS provides a more robust estimation of the period and amplitude of gapped time series, while the FFT's perfor-

mance is simultaneously dependent on the GP and frequency of the signal.




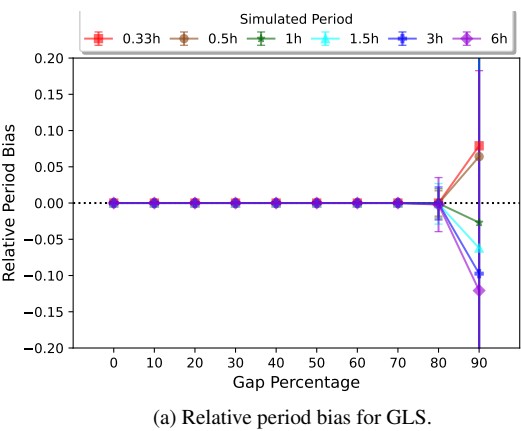
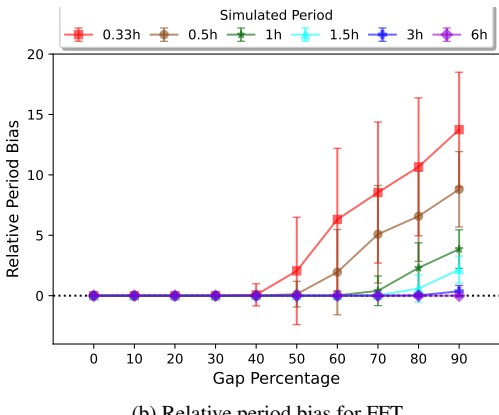

(a) Relative period bias for GLS.     (b) Relative period bias for FFT.

**Figure 2.** Comparison of relative period bias (Eq.6) as a function of gap percentage for each method, here shown for each simulated period (frequency). The bias is estimated from the average estimated values of the periods, and its standard deviation is scaled accordingly. Note that the y-axis (relative period bias) is limited between$[-0.2, 0.2]$ for the GLS and $[-4, 20]$ for FFT. This shows how extremely different the accuracy of each method is.

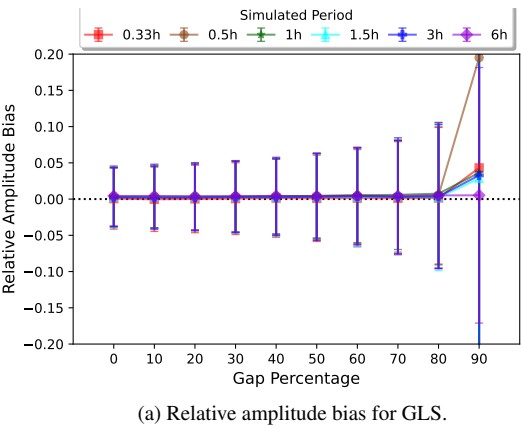
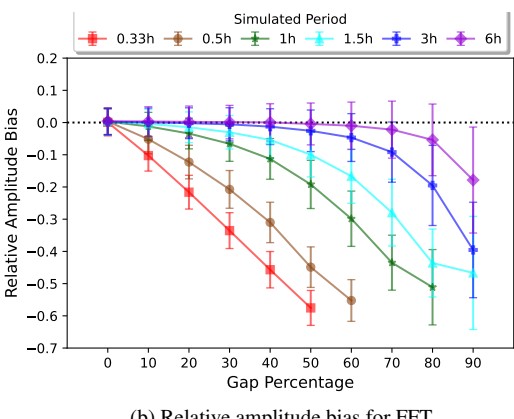

(a) Relative amplitude bias for GLS.     (b) Relative amplitude bias for FFT.

**Figure 3.** Comparison of relative amplitude bias (Eq. 6) as a function of gap percentage for each method, here shown for each simulated period (frequency). According to Fig.2b, past a certain GP threshold, the highest peak in the FFT spectrum does not belong to the true frequency but to meaningless interpolation noise. For this reason, the amplitude bias reported in Fig.3 is limited by relative period bias $< 3$, since amplitude values beyond this threshold should not be relied upon. Note that the upper and lower limits of the y-axis are different for each method as well.

## 5.2 Spectral Power Law Signal

A time series example is shown in Fig.4 to illustrate the complexity of a signal produced according to Sect. 3.2 for $\beta = 2$, showcasing the signal before and after the introduction of gaps. As the percentage of gaps in the data increases, the impact on
the spectral components varies depending on their frequency range. High-frequency components (rapid fluctuations over short periods) are most vulnerable when data is being removed. Subsequently, the lower frequency components (slow variations over





longer periods) follow, exhibiting greater resilience to data gaps. In essence, a considerably greater number of gaps is required to significantly affect the estimation of the latter. A distorted spectrum of this time series can be seen, which is a result of the limited sample length and resolution (Roberts et al., 1987). In addition, as the signal comprises numerous sinusoids with

random frequencies, the presence of closely spaced frequencies leads to the emergence of complex and broad peaks in the spectrum (Horne and Baliunas, 1986; Dewan and Grossbard, 2000).

In the absence of gaps, the true value of $\beta$ is accurately estimated from the spectra of all methods (see Fig.4a). However, after

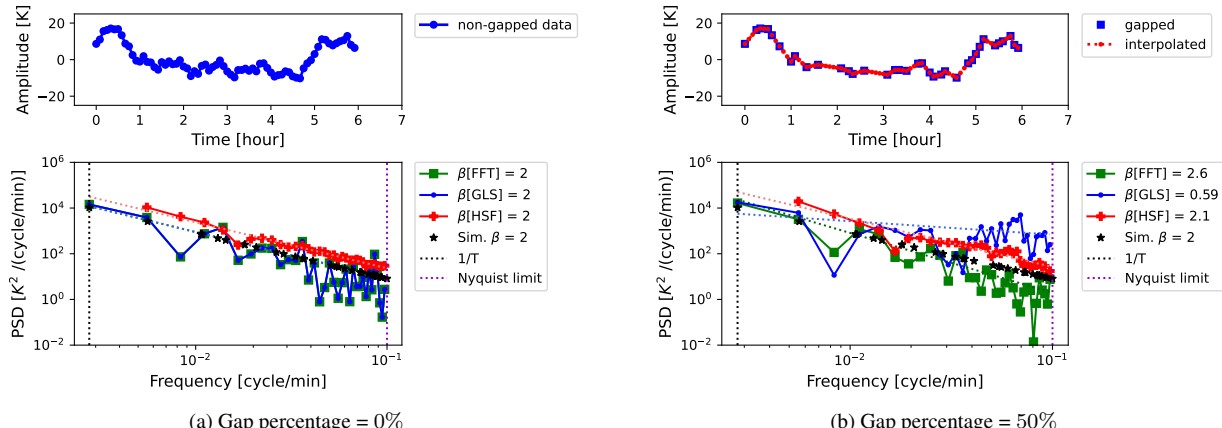

**Figure 4.** A time-series example (upper left and right) generated by the spectral power law simulation (according to Sect.3.2) with a spectral exponent of $\beta = 2$ within a 6-hour observation time, before and after the addition of gaps, respectively. The estimated power spectral densities of both non-gapped and gapped time series are shown in the lower left and right. Here, both x- and y-axes in the spectra figures were log-scaled so that a linear function can be identified. The dotted lines (in red, green and blue) represent the fits of the PSD of each method.

removing $50\%$ of the data points (Fig.4b), the estimated spectrum by the HSF remains relatively unaltered while the GLS and HSF spectra diverge. The overestimation of $\beta$ (bias$= 0.6$) by FFT is due to the amplitudes of high-frequency components being

underestimated, which is a result of the interpolation (Schulz and Stattegger, 1997; Hall and Aso, 1999). An even smaller bias (0.1) results in the case of the HSF. In contrast, the true power law can be seen for the first few low frequencies in the GLS spectrum, then it starts to flatten at intermediate and high frequencies with a substantial bias of $-1.41$. This occurs because the lack of data points constrains the least squares fit by GLS, which leads to the interpolation of power at these frequencies (i.e., a flat line).

For a statistically significant picture, we show the distributions of estimated $\beta$ values from 1000 simulation runs in Fig.5. In the non-gapped case, the distributions of all methods overlap within a small standard-deviation range of $\pm 0.2$ around $\beta = 2$. In contrast, we see that gaps cause the estimated $\beta$ values from the FFT and HSF spectra to spread over a larger range, while the distribution of GLS diverts far below the expected value of 2. The mean of estimated $\beta$ values from HSF spectra is clearly the closest to the true value. These distributions show that even in the absence of gaps, estimated $\beta$ lies mostly within the range of

$[1.5, 2.5]$ and not exactly at 2, as single spectra are distorted without averaging. It is also important to mention that the results of the bias are quite identical whether $\beta$ is estimated from averaging power-law exponents of single spectra or it is estimated



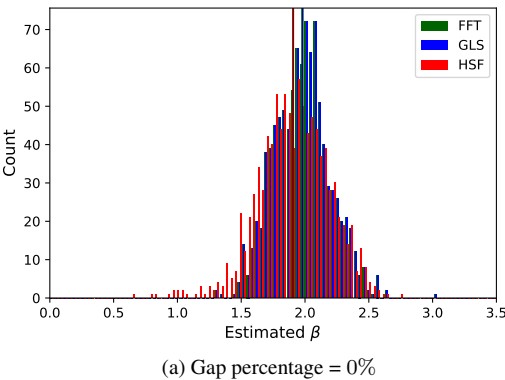
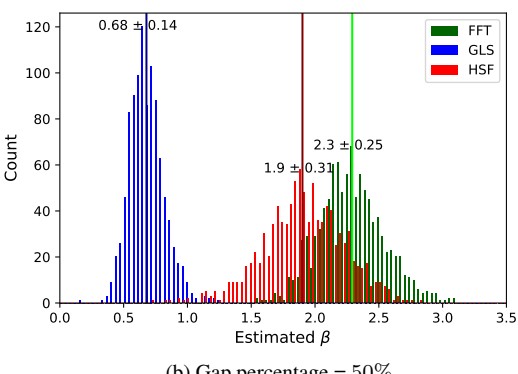

(a) Gap percentage = 0%       (b) Gap percentage = 50%

**Figure 5.** Histograms of the estimated $\beta$ values from 1000 spectra of time series generated by the spectral power law simulation (according to Sect.3.2) with a spectral exponent of $\beta = 2$ within a 6-hour observation time. The vertical lines in the histograms of the gapped case refer to the mean estimated values of $\beta$ by each method.

from fitting an averaged spectrum.

To further explore the behaviour of the bias under different conditions, we evaluated the effect of changing the simulated value of $\beta$ on the estimation bias (see Fig.6). At $0\%$ GP, all methods show no bias, except for $\beta > 2$ and $\beta = -1$, where there is an
apparent deviation. The first deviation is expected because the larger $\beta$ is than 2, the more the spectrum suffers from "low-frequency leakage" due to the finite length of the time series (Klis, 1994; Schulz and Mudelsee, 2002). The other case of small deviation takes place on the opposite end of the spectral slope range ($\beta = -1$), where high frequencies dominate. The power at these frequencies is quite easily aliased, hence, underestimated, as they are the closest to the Nyquist limit. These deviations do not mean that spectra of $\beta = -1$ or $\beta > 2$ can not be obtained, but that on average, they are very likely to be misestimated.
In the instance of gapped time series, our results show that as the GP increases, the biases in the estimated exponent also became more pronounced for all three methods. Similar to the non-gapped situation, the GLS demonstrates an exceptional efficiency in estimating flat spectra where $\beta = 0$ with a negligible bias. This is a consequence of the absence of frequency dependency in a flat spectrum, which renders the gaps irrelevant in terms of introducing bias, since the GLS spectrum is already flat. As $\beta$ increases (indicating a steeper decline in power with increasing frequency) and the percentage of gaps in the data increases,
the bias in the GLS spectrum becomes more prominent. For instance, in the case of $\beta$ transitioning from $1 \rightarrow 3$ where power is skewed towards low frequencies, gaps cause GLS to mistakenly assign excessive power at the missing high frequencies, ultimately resulting in a steady underestimation of $\beta$. In contrast, when $\beta = -1$, high-frequency components dominate low-frequency ones. Consequently, there is a loss of power at these high frequencies as the gaps disrupt their sampling. This loss of power causes the GLS method to overestimate $\beta$, mirroring the GLS bias observed when $\beta = 1$.
In similar fashion, both the FFT and HSF demonstrate a relatively constant bias for $\beta = 3$ of approximately $-0.3$ at all GPs. However, as $\beta$ decreases from $2 \rightarrow -1$, their biases monotonically increase as the GP increases. Nevertheless, the HSF shows substantially less bias than the FFT when the GP exceeds $10\%$. The FFT bias is attributed to the established interpolation





effects. Therefore, as more data points are interpolated, the FFT spectrum progressively underestimates the amplitude of the high frequencies. This underestimation results in the bias being positive for all $\beta$ values, except $\beta = 3$ where leakage causes

FFT to overestimate these frequencies. Overall, averaging $\beta$ values from single spectra is a good measure of the expected value because of their low standard-deviation except at very high GPs.

In light of the aforementioned considerations, it can be argued that the FFT technique demonstrates competence in generating accurate spectral estimations for non-gapped time series. Nevertheless, it encounters challenges as the data incorporates an increasing number of gaps, necessitating interpolation techniques which introduce inherent biases. Meanwhile, the HSF is

demonstrated to be a particularly reliable approach for analysing GW time series with spectral power law exponents $\beta \in \{1, 5/3, 2, 2.5, 3\}$ and in-between. Its performance, however, exhibits limitations primarily in cases where the spectrum of a time series possesses a power law exponent $\beta < 1$. Notably, such occurrences have only been observed and predicted within measurements of vertical wind time series, as indicated in Tab.A1. One can reasonably anticipate that spectra falling within the range of $\beta$ values between 1 and 3 will be prevalent across the majority of atmospheric time series.

Conversely, the GLS method yields similarly favourable outcomes, particularly for time series whose spectra are flat and high-frequency dominated, that it even surpasses the accuracy of HSF when $\beta$ possesses values between $-1$ and $0$. Nonetheless, the GLS method exhibits an increasing bias as the value of $\beta$ increases beyond $0$, rendering it a less certain choice for time scales extending beyond a few hours, which commonly occur in the context of atmospheric gravity waves. Clearly, the consistent and overall impeccable GLS performance on estimating periods and amplitudes of single sinusoids does not seem to translate to

universally resolve the level of superposition of many random periodicities with power-law amplitudes.

## 6   Discussion

### 6.1   Low-Frequency Leakage

The problem of power leakage from low frequencies into higher ones arises as a result of the constrained frequency range,

which of itself is limited by the observed time span $T$. This leakage does not only take place in the case of spectra with $\beta > 2$, but also in the spectra of time series with scales longer than $T$. GWs can often have these kinds of scales with periods longer than the simulated, $6\,\mathrm{h}$ and normally these periods are not resolved. Thus, we also quantitatively tested the effect of these long periods on the estimated spectra by adding 3 extra waves of periods 8, 10, and $12\,\mathrm{h}$ into the simulated time series. Two cases were examined, one where each of the 3 waves has an amplitude equivalent to the lowest frequency component in each

simulation. In another case, we scaled their amplitudes by the same power law exponent $\beta$ as all frequencies in the simulation. In both cases, longer-than-$T$ periods produced quite similar effects on the spectra, with a substantial positive bias observed for all signals with $\beta < 2$ and a significant negative bias for $\beta > 2$, even in the absence of gaps. For instance, Figs. 7a,7b show an example of this leakage in averaged spectra for $\beta = 3$, without and with the extra waves. A common feature of both cases is the spectral power being excessively concentrated at the lowest frequencies. When the extra long waves are added, the leakage

becomes more drastic for the GLS and FFT, and their absolute biases of $\beta$ increase. In contrast, the HSF is less affected by these



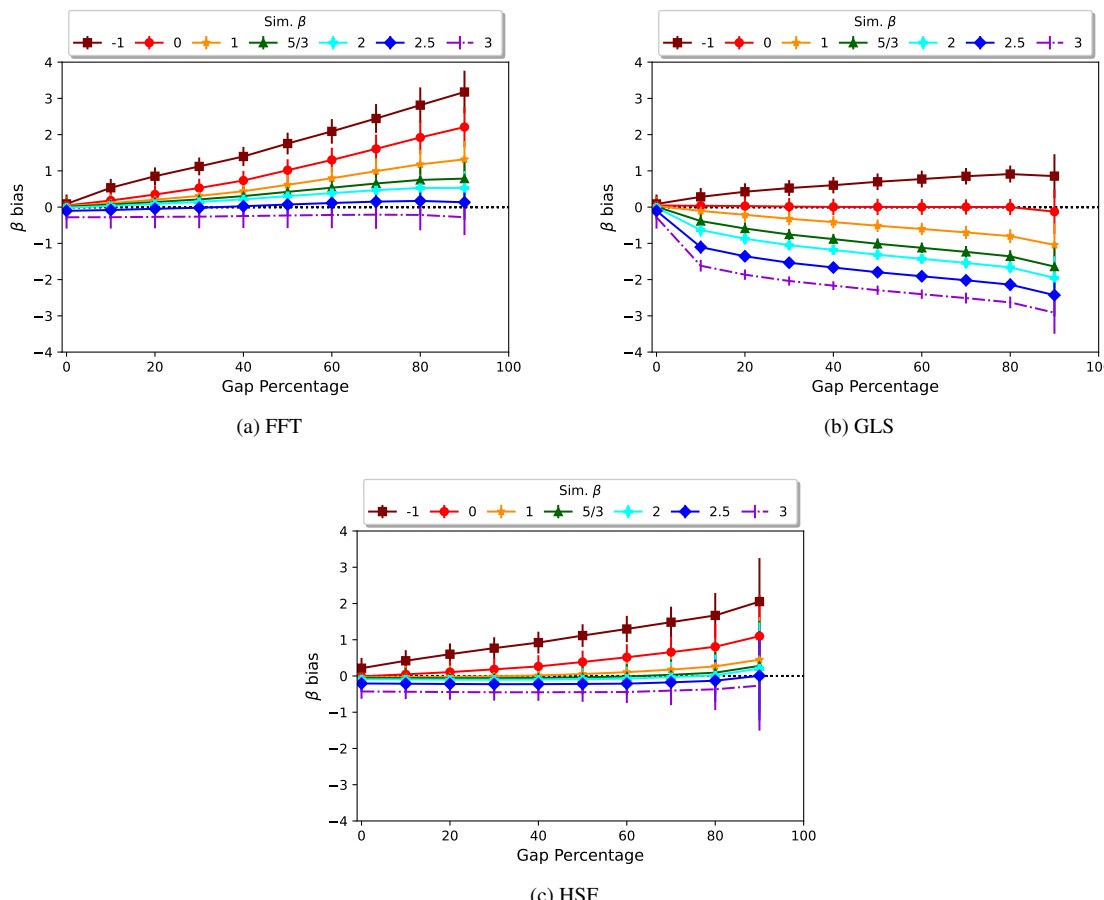

**Figure 6.** Comparison of the bias in the mean $\beta$ estimates obtained from the three different methods as a function of gap percentage. The results of each method are shown for spectra with power law exponents $\beta \in \{-1, 0, 1, 5/3, 2, 2.5, 3\}$

long periods compared to the FFT and GLS. This effect of longer-than-$T$ waves in the time series resembles that of trends, which contribute to the spectral shape with a power-law exponent $\beta = 2$ (Klis, 1994). While a weighted fit of the spectra can reduce this bias, it does not fully rectify the problem of leakage, it also requires a smoothed spectrum and may be confounded by other biases from observational noise, gaps, or method inefficiencies.

To counteract this leakage, we test the approach of "prewhitening" the data followed by "postdarkening" the spectra. It is a technique to decorrelate the time series before calculating the PSD, which was coined by Blackman and Tukey (Blackman and Tukey, 1958) and mentioned throughout literature (Dewan and Grossbard, 2000; Guharay and Sekar, 2011). To apply prewhitening, we first-difference the time series, i.e. subtract each data point from its previous value. After obtaining the power spectrum, it is first smoothed using a Hann window to reduce the fluctuations, making it easier to fit the spectral shape. Then 330 the spectrum is postdarkened through division by a factor of $2(1 - \cos(2\pi f_n \Delta t))$, which compensates for prewhitening (i.e., recolours) the time series.




In Fig. 7c, we present the postdarkened spectra after prewhitening the time series for $\beta = 3$. This approach completely cancels out the bias in all methods for both cases. This confirms the effectiveness of the prewhitening and postdarkening method in correcting the leakage problem. However, this approach is not a perfect solution since it may introduce additional bias for less steep spectra (where $\beta < 1$) which do not suffer from leakage.

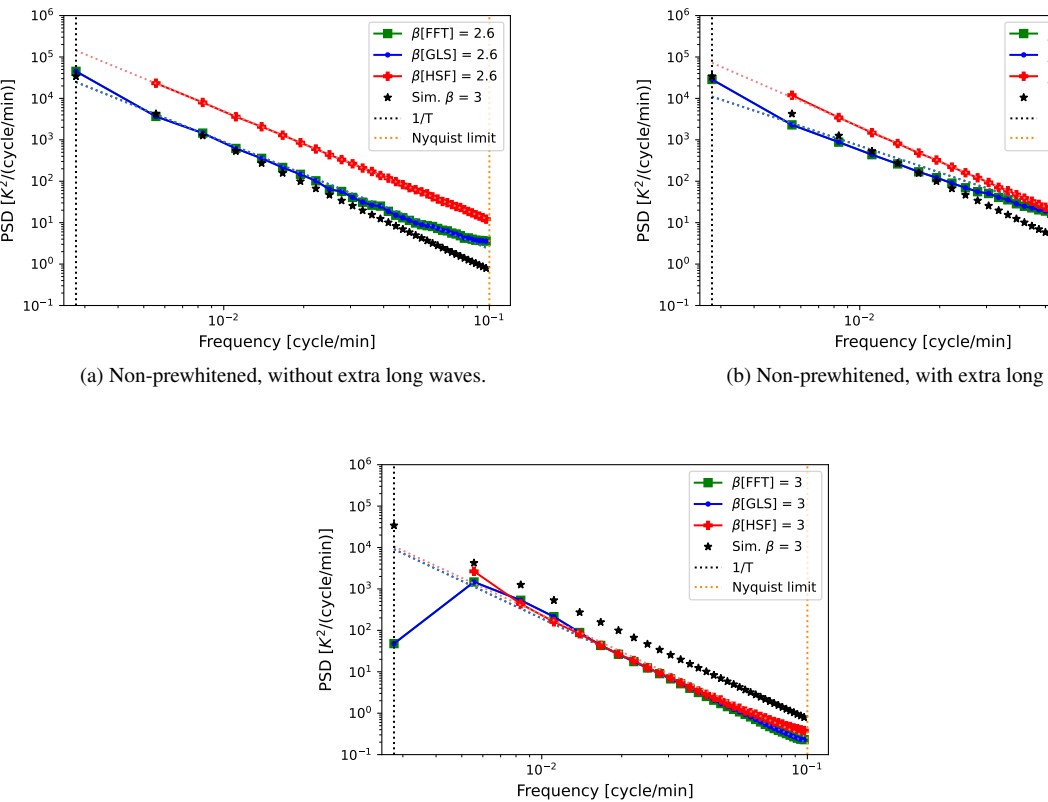

**Figure 7.** Averaged temporal spectra of non-gapped time series generated by the spectral power law simulation with a spectral exponent of $\beta = 3$ within a 6-hour observation time (a) without extra waves, (b) after the addition of 3 extra waves with frequencies lower than $f = 1/T$ (particularly 8, 10, and 12 hours) to the simulation, (c) the postdarkened spectra of the prewhitened time series with extra waves. Here, also both axes were log-scaled so that a linear function can be identified. The dotted lines (in red, green and blue) represent the fits of the PSD of each method.

## 6.2 Method Selection Procedure

The spectral analysis of GW time series data is a complex task that requires careful consideration of various factors. Based on our simulation results, we propose a flowchart (see Fig. 8) that outlines a practical guide for selecting appropriate spectral estimation methods for GW studies, taking into account the characteristics of the observed data such as its complexity and



percentage of gaps in it. From the flowchart, it is clear that there are recognisable differences between the patterns of the time series of superposed scales and single sinusoids. Even within the former classification, the anti-persistent time series whose spectra have $\beta \in [0, -1]$ are still differentiable from those long-range dependent ones where $\beta \in [1, 3]$. It is also safe to say that theoretical predictions of GW spectra with $\beta \in [0, -1]$ exist only for measurements of vertical wind time series (see Tab. A1).

Otherwise, $\beta \in [1, 3]$ spectra should be expected for the vast majority of atmospheric time series. Note that, even in the absence of gaps in the signal, caution must be taken if the estimated $\beta$ is approximately equal to or less than 2. If it is, then it can be an accurate estimation, or caused by one of the systematic errors such as interpolation or longer than $T$ variations.

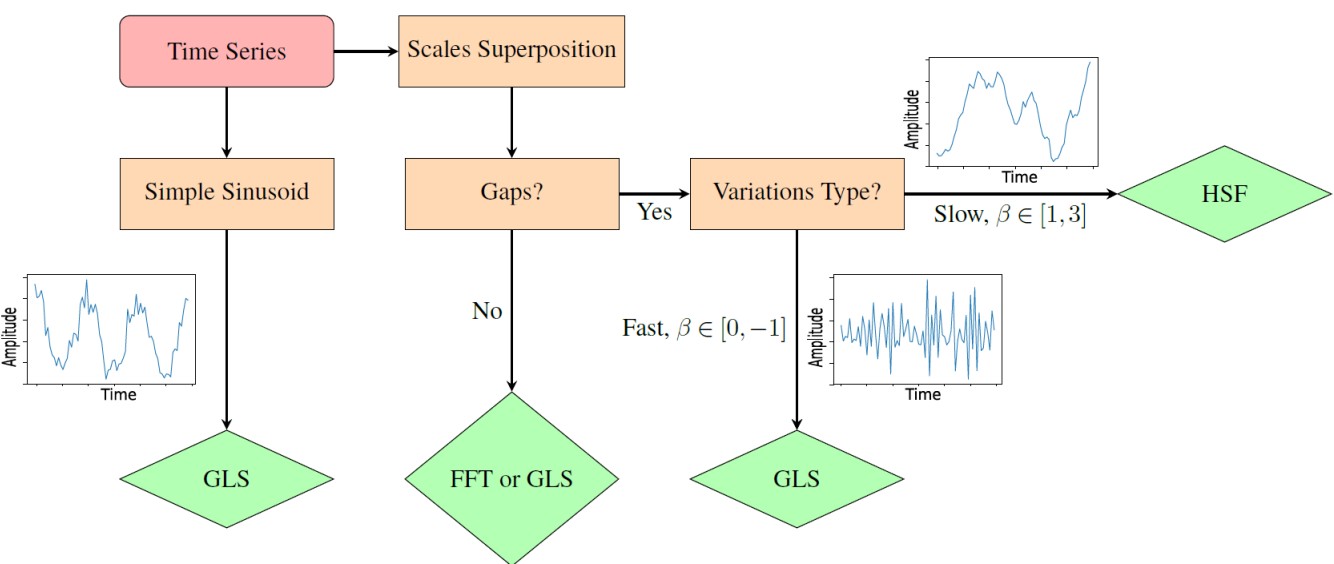

**Figure 8.** Recommended procedure for estimating power spectra of gravity waves time series.

## 7 Conclusions

Our study investigated the bias of the methods when estimating the spectra of GW time series using simulated data with various characteristics. We first examined simulated signals consisting of single sinusoids in order to characterise real quasi-monochromatic waves. The main findings are:

– The GLS and FFT are identical in the case of evenly sampled data with no gaps.

– The GLS has relatively negligible bias in estimating the frequencies and amplitudes of gapped single sinusoids compared
to the FFT.




- Up to $80\%$ GP, the performance of the GLS shows no dependency on the frequency of the signal and proves efficiency in mitigating the effects of gaps in the data, while the absolute bias of the FFT drastically increases as both the GP and frequency increase.

- The inefficiency of the FFT in recovering the true frequency and amplitude of the signal is attributed to the required interpolation step, which distorts the time series.

- Overall, the GLS is a more reliable method for identifying the periodic components of gapped GW time series, particularly when dealing with high-frequency sinusoids.

Then we investigated the effects of gaps on the spectra of simulated time series composed of a superposition of scales with power-law amplitudes. Our results are summarised as follows:

- The choice of spectral estimation methods can significantly impact the accuracy of $\beta$ estimation, especially in the presence of data gaps and leakage effects.

- The FFT is competent in generating accurate spectral estimations for non-gapped time series but faces challenges with increasing GP as it is interpolation-based, which is a source of biases.

- The HSF is reliable for analysing GW time series with spectral power law exponents $\beta \in [1,3]$, but exhibits limitations when $\beta$ is less than $1$.

- The GLS method performs quite well for time series with flat and high-frequency dominated spectra, surpassing the accuracy of HSF for $\beta \in [-1,0]$, but exhibits increasing bias for exponents beyond $0$, making it less suitable for longer time scales.

- The reliance on an approach that is not interpolation-based, proves more advantageous in mitigating such biases.

- Even in the absence of gaps, all tested methods underestimate $\beta$ for time series whose spectra are too steep ($\beta > 2$), due to the leakage of power at low frequencies into high-frequency components.

- Waves that are longer than the observed time span can introduce similar leakage biases in $\beta$ estimation, with positive biases for $\beta < 2$ and negative biases for $\beta > 2$.

- The HSF was less affected by these slow variations compared to the FFT and GLS.

- Prewhitening the time series followed by postdarkening the spectra is recommended as a suitable approach to correct these low-frequency leakage problems.

Our findings highlight the importance of carefully selecting appropriate spectral estimation methods and accounting for potential biases caused by data gaps, leakage effects, and long periods when interpreting $\beta$ values from GW observations. Our findings contribute to the understanding of the limitations and uncertainties associated with $\beta$ estimation and provide guidance



for future research and advancements in spectral analysis techniques to improve the accuracy and reliability of $\beta$ estimation in
GW studies and to better understand the physical processes driving GW variability in the atmosphere.

*Code availability.*  The code to simulate time series with power-law spectra, analyze and fit them is accessible under (Mossad, 2023).

**Table A1.** Comparison of theoretical predictions and selected observed values of the power-law exponent $\beta$ of GW spectra. Here $T$ refers to temperature, $W$ is wind and $\rho$ is density.

| Reference | Type of Spectra | Spectral exponent or $\beta$ |
|---|---|---|
| Universal spectrum (VanZandt, 1982) | Vertical wavenumber spectra by Doppler navigator and anemometer of *horizontal W* observations | 2.4 |
| Linear instability theory (Dewan and Good, 1986; Smith et al., 1987) | Vertical wavenumber spectra of *horizontal W* | 3 |
| Saturated-cascade theory (Dewan, 1994) | Horizontal wavenumber spectra of *horizontal W*, *T* and fractional $\rho$ | 5/3 |
| Saturated-cascade theory (Dewan, 1994) | Vertical wavenumber spectra of *vertical W* | -1 |
| Saturated-cascade theory (Dewan, 1994) | Temporal spectra of *vertical W* | 0 |
| Saturated-cascade theory (Dewan, 1994) | Temporal spectra of *horizontal W*, *T* and fractional $\rho$ | 2 |
| Lidar observations (Shibata et al., 1988) | Vertical wavenumber spectra of *T* data | 2.5 to 3 |
| Diffusive filtering theory (Gardner, 1994) | Vertical wavenumber spectra of *horizontal W* | 3 (p=2) |
| Diffusive filtering theory (Gardner, 1994) | Temporal spectra of *horizontal W* | 2 (p=2) |
| Diffusive filtering theory (Gardner, 1994) | Vertical wavenumber spectra of *vertical W* | -1 (p=2) |
| Diffusive filtering theory (Gardner, 1994) | Temporal spectra of *vertical W* | 0 |
| Diffusive damping theory (Weinstock, 1990; Zhu, 1994) | Temporal spectra of *horizontal W* | p |
| Diffusive damping theory (Weinstock, 1990; Zhu, 1994) | Vertical wavenumber spectra of *horizontal W* | 3 |
| Doppler spread theory (Hines, 1991) | Vertical wavenumber spectra of *horizontal W* | 3 |
| Doppler spread theory (Hines, 1991) | Temporal spectra of *horizontal W* | p |
| Lidar observations (Gardner et al., 1995) | Temporal spectra of $\rho$ data | 2.3 |
| Lidar observations (Gardner et al., 1995) | Temporal spectra of *T* data | 1.6 |
| Lidar observations (Gardner et al., 1995) | Temporal spectra of *vertical W* data | $\approx 0$ |
| Lidar observations (Gardner et al., 1995) | Vertical wavenumber spectra of $\rho$ data | 3.5 |
| Lidar observations (Gardner et al., 1995) | Vertical wavenumber spectra of *T* data | 2.5 |
| Lidar observations (Gardner et al., 1995) | Vertical wavenumber spectra of *vertical W* data | 1.4 to 1.9 |
| Radiosonde observations (Zhang et al., 2017) | Vertical wavenumber spectra of *zonal W* data | 2.4 to 2.68 |
| Radiosonde observations (Zhang et al., 2017) | Vertical wavenumber spectra of *meridional W* data | 2.53 to 2.76 |
| Radiosonde observations (Zhang et al., 2017) | Vertical wavenumber spectra of *vertical W* data | 0.2 to 0.3 |
| Balloon observations (He et al., 2020) | Vertical wavenumber spectra of *T* data | 2.18 to 2.63 |



*Author contributions.* M.M. wrote the codes to conduct these simulations, analyzed their results and drafted the manuscript. I.S., R.W., and G.B. provided supervision, scientific insight, and edited the text of the manuscript.

*Competing interests.* Robin Wing and Gerd Baumgarten are members of the editorial board of Atmospheric Measurement Techniques.

*Acknowledgements.* This paper is a contribution to the project W1 (Gravity Wave Parameterization for the Atmosphere) of the Collaborative Research Centre TRR 181 "Energy Transfers in Atmosphere and Ocean" funded by the Deutsche Forschungsgemeinschaft (DFG, German Research Foundation) - Projektnummer 274762653 and the Analyzing the Motion of the Middle Atmosphere Using Nighttime RMR-lidar Observations at the Midlatitude Station Kühlungsborn (AMUN) funded by Deutsche Forschungsgemeinschaft (DFG) - Projektnummer
395 445400792.



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
