# Peer review of "Assessing Atmospheric Gravity Wave Spectra in the Presence of"

_EGUsphere, 2023_

## Referee Comment (RC2)

Comments on Assessing Atmospheric GravityWave Spectra in the Presence of Observational Gaps

Mohamed Mossad, Irina Strelnikova, Robin Wing, and Gerd Baumgarten

**General comments:**

Limitations of spectra:

There has been a resurgence on interest in techniques for determining scaling exponents, scaling properties of geosystems. Over the last decades many techniques have been developed, particularly to characterize/quantify the intermittency/ multifractal aspects of the system (e.g. trace moments, double trace moments, generalized structure functions etc.). Although the atmosphere is highly turbulent/intermittent/multifractal, the effect of the corrections is often viewed – as it was 50 years ago - as no more than an often fairly small "intermittency correction" to the spectral exponent (the $K(2)$ in this paper). This is a shame since even if this correction to quasi-Gaussian spectral exponents is small (for the velocity field it is typically of the order of 0.15 in the horizontal, but closer to 0.25 in the vertical), the effect of intermittency may nevertheless be huge if measured for example at moments higher than 2 (the moment relevant for the spectrum). For example, the physically important energy fluxes depend on the 3rd moment of the velocity and their variances depend on the 6th moment, and the latter are apparently divergent (see the review [1])!

Overall, the focus on spectra – that at best characterize the second moment – is thus rather limiting, especially if one is attempting to test theories that predict $H$ (typically on dimensional grounds), but that ignore the intermittency. In these cases, at best the spectra must be supplemented with other techniques to determine $K(2)$ so that $H$ can be determined using the relation $\beta = 1+2H - K(2)$. Techniques such as Haar fluctuations that directly estimate $H$ are therefore advantageous.

Difficulty in estimating spectra when data are missing:

An additional problem with spectra is that they are notoriously difficult to estimate when there are missing data. For example, using interpolation to fill gaps (one of the methods used here) can cause huge biases. This is because the spectral exponent is related to $H$ which is the maximum order of differentiability of the series (it is related to the fractal dimension of the signal). Therefore - as is typically the case in geophysics where $H<1$ then using linear interpolation (i.e. using nonfractal curves with $H = 1$) will badly bias the statistics (depending on the amount of missing data). This should be stated somewhere in this paper. Alternatively, as discussed to some extent by the authors if instead of interpolation, Lomb-Scargle is used, one finds that it has big problems with spectral leakage [2] and these are often not improved with Multi-Taper Methods (as advocated by some [3]). This could be mentioned.

Gravity waves

Since the atmosphere is highly nonlinear / turbulent, with Reynolds number of the order of $10^{12}$, it is not clear why linear gravity wave theory should apply. [4] have argued that in reality, what is observed is a consequence of (high Reynold's number) scaling, intermittent fractional wave equation consistent with stratified Kolomogorov type turbulence. This would explain the existence of wave-like structures in the high Reynold's number limit as well as the observed exponents and their intermittencies. This alternative possibility should be mentioned.

**Specific comments:**

The paper is essentially a series of numerical tests of analysis techniques applied to synthetic series with a gap model, I have problems with both.

First the simulation method is problematic since it is not clear what the statisitcs of the resulting series are. Since it is a linear model, it should be nonintermittent (monofractal), and therefore presumably quasi-Gaussian, but this needs clarification. It is not trivial to even theoretically deduce the spectrum from the mathematical definition. Even if one wants to avoid more realistic multifractal models and stick to Gaussian ones, why not use fractional Brownian motion of fractional Gaussian noise that are standard, well-defined processes? At least we would know exactly what we are dealing with in the absence of data gaps and analysis issues!

Also of concern is the gap model. The statistics of the model are perhaps not as clear as they seem. The problem is that when the probability of gaps becomes large enough, many join together to become "supergaps", ultimately the left-over nongaps - where the data are sampled – may be a fractal set, see Mandelbrot's "trema" constructions (and theory) in [5]. The thing is that the fractal dimension then becomes a crucial characteristic of the sampling, and thus of the biases.

Sampling data with fractal holes/gaps is indeed highly pertinent since it seems to be a fairly general problem with many geophysical data sets. However it requires its own study: how does the fractal dimension of the sampling affect the analyses?

**Haar:**

I was disappointed that the Haar technique was not further discussed since it is the only one that (largely) avoids most the problems discussed above (I did note that the biases reported in the paper are apparently the smallest for the Haar, as found already in [6] and reiterated in [2]). Although it was mentioned that it is easy to apply to data with gaps, the reference to this nonuniform algorithm should be given (it is in appendix of [7]). The Haar method also allow for the determination of the intermittency corrections (the entire codimension function in fact).

**References**

1. Lovejoy, S., *Scaling, dynamical regimes and stratification: How long does weather last? How big is a cloud? .* Nonlinear Processes in Geophysics 2023. **30**: p. 311–374.
2. Hébert, R., K. Rehfeld, and T. Laepple, *Comparing estimation techniques for temporal scaling in palaeoclimate time series.* Nonlin. Processes Geophys., 2021. **28**: p. 311–328.
3. Springford, A., G.M. Eadie, and D.J. Thomson, *Improving the Lomb–Scargle Periodogram with the Thomson Multitaper. .* The Astronomical Journal,, 2020. **159** p. 205.
4. Pinel, J. and S. Lovejoy, *Atmospheric waves as scaling, turbulent phenomena.* Atmos. Chem. Phys. , 2014. **14**: p. 3195–3210.
5. Mandelbrot, B.B., *The Fractal Geometry of Nature*. 1982, San Francisco: Freeman.
6. Lovejoy, S. and D. Schertzer, *Haar wavelets, fluctuations and structure functions: convenient choices for geophysics.* Nonlinear Proc. Geophys. , 2012. **19**: p. 1-14.
7. Lovejoy, S., *A voyage through scales, a missing quadrillion and why the climate is not what you expect.* Climate Dyn., 2015. **44**: p. 3187-3210.

---

## Author Response (AR1)

**Response to Reviewer #1**

*We appreciate the comments and references given by both reviewers.*
*We repeat the reviewers' concerns and provide our respective responses in italics. The changes are in the revised manuscript.*

The manuscript describes a thorough comparison of different methods to quantify spectral characteristics of geophysical one-dimensional data such as profiles or time series. The motivation comes from analyses of gravity waves in observations. An important focus is on the effect of potential gaps in the data, with gaps present for a significant fraction of the data. Three methods are compared: Fast Fourier Transform, Generalised Lomb-Scargle periodogram, and Haar Structure Function. The study constitutes a useful and interesting contribution to the literature: the topic is rather technical and only interests a specific readership, but this is very well carried out, well explained and generally well presented. A very positive aspect is that an open code is made available to use these three analysis methods. Some changes would constitute improvements, and in particular, the conclusion section deserves to be rewritten in less technical terms. Minor revision is recommended, and this shall constitute a valuable study, although for a somewhat specific audience. **Main concern:**
The authors have made appreciated efforts for clarity throughout the text and for pedagogy in presenting the different methods. This is particularly useful as these methods are not necessarily familiar to all, making this study a valuable opening towards uncommon methods that may be of relevance for certain purposes (data with significant gaps in particular).

However, the conclusion appears less polished than the rest of the text: a list of key messages has been identified, and this is positive and useful, but this list remains too technical and too long. It makes the conclusion less readable and less efficient than it could be. There should be more of an effort to come back to recommendations for concrete analyses of observations, with less acronyms and simpler messages and recommendations for the different situations and the advantages/disadvantages of the different methods. There is a contrast between the conclusion, which reads more like a summary of a technical report, and the synthetic sketch of Figure 8, which carries simple and clear messages.

*As requested, we rewrote the conclusion section and addressed the mentioned points. We hope that the text flows more smoothly.*

**Minor points:**
l30 'universal GW spectrum': odd formulation given that different power laws have been documented in different contexts as recalled in table A1

*We have removed the word "universal" in l30.*

Additionally, although the table does not aim to be exhaustive, it could include observations

from long-duration balloons, as these provide original insights into Lagrangian spectra of gravity waves, which is uncommon (Hertzog Vial 2001, Podglajen et al. 2016)

*Thanks for the references, we have cited the two papers (Hertzog Vial 2001, Podglajen et al. 2016) in Table A1.*

l50-51: the FFT is likely known by almost anyone doing data analysis; however, how common are the other two methods? How commonly are they used in geosciences? Could the authors give some hints or suggestions on that?

*The following text is rewritten in l50-51:*
*"The FFT is the standard method to analyse spectra of evenly sampled data. The Lomb-Scargle periodogram (LS) was used in many studies as the main analysis method (or as a reference) of GW spectra (e.g., Hall and Aso, 1999; Zhang et al., 2006; Guharay and Sekar, 2011; Qing et al., 2014). As far as the authors are aware, the HSF has never been used in atmospheric GW studies. Both the GLS and HSF are specifically known to handle unevenly sampled data."*

l129: it is very good that there is an open code available for part of these tools

*Thank you.*

l131: the study may be motivated by the analysis of gravity waves, but the conclusions and the methods described are more general. Signals are analyzed for their periodic components or for their spectral slopes, but no use is made of specific aspects of gravity waves such as polarization relations. Other scientists dealing with observations including gaps, and analyzing nearly periodic signals and/or spectra, could be very interested in this. The authors are thus encouraged to put less stress on the gravity wave aspect and to broaden the scope of the study (by a few appropriate sentences, in the introduction, for instance).

*The following text is added to introduction l49:*
*"Even though this study is motivated by the analysis of atmospheric GWs, the conclusions and the methods can be generalised to different fields with similar time series characteristics such as astronomy and seismology."*

l162: it could be recalled that in many observational cases, the f is not a frequency but a vertical wavenumber.

*The following text is added to the simulation section 3.2:*
*"(in the case of spatial data, PSD $\propto 1/(k, l, m)^\beta$ where $k, l, m$ are horizontal and vertical wavenumbers)."*

*In addition, our methods are general enough to also be used on vertical lidar profiles. For example, in the case of a vertical lidar profile with a resolution of 0.5 km, this would correspond to a 36 km measurement range. This means in our simulation, vertical wavelength would be in range of 1 km to 36 km.*

l204: a reference to the MLE could be included

*(Duvall and Harvey, 1986) citation moved to l204.*

l205: why is the observation O noted as a vector?

*Thank you, we removed it.*

Figure 5: in the right panel, the figure includes the mean and standard deviation. This should be added in the left panel. The comparison for HSF is particularly important.

*Figure 5 has been updated.*

l325-331: prewhitening and postdarkening should be explained a bit more. How does this relate to derivation and integration? Can the factor on line 330 be explained or interpreted in a few words?

*Spectral leakage is a well-known problem, there are different methods to deal with it. We re-explained the prewhitening and postdarkening part in the discussion and gave a more inclusive overview without going into much details in order not to confuse the reader, since it is a vast topic.*

Figure 8: rather than "simple sinusoid", which describes the synthetic data used for the study, the authors should find a phrasing that could better describe potential observations ("conspicuous periodic signal"? "signal with one dominant frequency"? ). How (un-)important is the sinusoidal character of the oscillations?

*The assumption of sinusoids is essential, especially for gravity waves in the absence of nonlinear effects. In addition, the FFT decomposes the signal into sines and cosines while the GLS fits a weighted (full) sine function to the signal (Eq.2). We changed it to a "signal with single/one frequency" in Figure 8 and throughout the manuscript.*

l350: the authors should take into account that readers may skip to the conclusion for the main messages: some redundancy between the conclusion and the preceding text is not a problem

and rather desirable if it makes the conclusion more self-consistent. For instance, it is worth recalling what "the methods" are.

*We redefined the methods in the conclusion section.*

l354: the editor may judge otherwise, but redefining acronyms could be welcome.

*We agree, we redefined all acronyms in the conclusion section.*

l365: recall that beta is the spectral slope

*We added that beta is the spectral slope.*

l367: 'competent', for a method? Efficient?

*Thank you, we changed it accordingly.*

l380: re-explain prewhitening and postdarkening, very briefly; part of the readers will be unfamiliar with these.

*We re-explained prewhitening and postdarkening briefly in the conclusion section.*

Hertzog, A., & Vial, F. (2001). A study of the dynamics of the equatorial lower stratosphere by use of ultra-long-duration balloons: 2. Gravity waves. Journal of Geophysical Research: Atmospheres, 106(D19), 22745-22761.

*Added citation*

Podglajen, A., Hertzog, A., Plougonven, R., & Legras, B. (2016). Lagrangian temperature and vertical velocity fluctuations due to gravity waves in the lower stratosphere. Geophysical Research Letters, 43(7), 3543-3553.

*Added citation*

**Response to Reviewer #2**

*We appreciate the comments and references given by both reviewers.*

*We repeat the reviewers' concerns and provide our respective responses in italics. The changes are in the revised manuscript.*

**Limitations of spectra:**
There has been a resurgence of interest in techniques for determining scaling exponents, and scaling properties of geosystems. Over the last decades, many techniques have been developed, particularly to characterize/quantify the intermittency/ multifractal aspects of the system (e.g. trace moments, double trace moments, generalized structure functions etc.). Although the atmosphere is highly turbulent/intermittent/multifractal, the effect of the corrections is often viewed – as it was 50 years ago - as no more than an often fairly small "intermittency correction" to the spectral exponent (the K(2) in this paper). This is a shame since even if this correction to quasi-Gaussian spectral exponents is small (for the velocity field it is typically of the order of 0.15 in the horizontal, but closer to 0.25 in the vertical), the effect of intermittency may nevertheless be huge if measured for example at moments higher than 2 (the moment relevant for the spectrum). For example, the physically important energy fluxes depend on the 3rd moment of the velocity and their variances depend on the 6th moment, and the latter are apparently divergent (see the review [1])! Overall, the focus on spectra – that at best characterize the second moment – is thus rather limiting, especially if one is attempting to test theories that predict H (typically on dimensional grounds), but that ignore the intermittency. In these cases, at best the spectra must be supplemented with other techniques to determine K(2) so that H can be determined using the relation b =1+2H - K(2). Techniques such as Haar fluctuations that directly estimate H are therefore advantageous.

*We added the following text to l123:*
*"Despite that the term K(2) is fairly small, it is nontrivial in the non-Gaussian case and higher moments q, and the HSF allows for its calculation (Lovejoy and Schertzer, 2012), but this is beyond the scope of this paper."*

*Based on 1000 simulation runs (see attached Figure), the estimated slopes (β=2) were averaged to estimate the ensemble statistics after accounting for K(2) at gap percentage 0%. We see that the estimated β distribution of the corrected Haar Structure Function is shifted away from the true slope by 0.08. Since we know the slope of the simulated spectra and know that we did not introduce any intermittency to the time series the corrections should be small, and do not affect the final conclusions. The mean estimated β from both FFT and GLS is 1.98 (see Figure 5 in the manuscript) the uncorrected HST is in slightly better agreement with expectations.*

[Figure]

*Additionally, the Haar Structure Function is already the least biased method for gapped data (without correction) and the approximation $K(2) \approx 0$ should be valid.*

**Difficulty in estimating spectra when data are missing:**

An additional problem with spectra is that they are notoriously difficult to estimate when there are missing data. For example, using interpolation to fill gaps (one of the methods used here) can cause huge biases. This is because the spectral exponent is related to H which is the maximum order of differentiability of the series (it is related to the fractal dimension of the signal).

Therefore - as is typically the case in geophysics where H<1 then using linear interpolation (i.e. using nonfractal curves with H = 1) will badly bias the statistics (depending on the amount of missing data). This should be stated somewhere in this paper.

*We used the linearly interpolated time series only for the FFT because this was commonly used, while the GLS and HSF were directly applied on gapped data without interpolation. We mentioned that FFT is biased due to linear interpolation in line 255 with references to the effect, and later in the text as well.*

*To further stress this point we have added the following text to l255:*

*"This overestimation of β can also be explained by the fact that these interpolated (high frequency) components contribute locally by β = 3 to the overall slope of the spectrum which result in positive bias when the true β is less than 3 (Lovejoy, 2014)."*

Alternatively, as discussed to some extent by the authors if instead of interpolation, Lomb-Scargle is used, one finds that it has big problems with spectral leakage [2] and these are often not improved with Multi-Taper Methods (as advocated by some [3]). This could be mentioned.

*We added the following text to l66:*

*"These three studies showed that the LS method suffers from significant leakage in the case of power-law spectra which persists even when Multi-Taper Methods (MTM) are used, however, a*

*combination of both the LS and MTM seems to improve on its disadvantages (Springford et al., 2020)."*

**Gravity waves**

Since the atmosphere is highly nonlinear/turbulent, with Reynolds number of the order of 1012, it is not clear why linear gravity wave theory should apply. [4] have argued that in reality, what is observed is a consequence of (high Reynold's number) scaling, intermittent fractional wave equation consistent with stratified Kolomogorov type turbulence. This would explain the existence of wave-like structures in the high Reynold's number limit as well as the observed exponents and their intermittencies. This alternative possibility should be mentioned.

*In the revised manuscript in l137, we have acknowledged the alternative possibility regarding high Reynolds number stratified turbulence in addition to other theories as follows:*

*"While our simulation adopts a simplified linear saturation theory approach through the superposition of sine waves (Dewan and Good, 1986; Smith et al., 1987), we acknowledge that other explanations for the spectral character of GWs exist, including "nonlinear-damping" (Weinstock, 1982, 1990; Gardner, 1994), "Doppler spreading" (Hines, 1991), "saturated-cascade similitude" (Dewan, 1994) and high-Reynolds-number "stratified turbulence" (Pinel and Lovejoy, 2014), see Table.A1 for more details."*

*We recognize that this perspective could offer a more comprehensive understanding of the observed spectral character of gravity waves.*

**Specific comments:**

The paper is essentially a series of numerical tests of analysis techniques applied to synthetic series with a gap model, I have problems with both. First the simulation method is problematic since it is not clear what the statistics of the resulting series are. Since it is a linear model, it should be nonintermittent (monofractal), and therefore presumably quasi-Gaussian, but this needs clarification. It is not trivial to even theoretically deduce the spectrum from the mathematical definition. Even if one wants to avoid more realistic multifractal models and stick to Gaussian ones, why not use fractional Brownian motion of fractional Gaussian noise that are standard, well-defined processes? At least we would know exactly what we are dealing with in the absence of data gaps and analysis issues!

*The waves that we observe on a given night of observations are not always well described by a standard distribution. Some nights have clearly dominant quasi-monochromatic waves (a reason to have the signal with one frequency simulation in Section 3.1), other nights have a more distributed set of waves. Differences exist in summer vs winter measurements. As well, the observed wave population changes as a function of altitude or in the presence of dominating geophysical flows and features such as the Polar Vortex.*

*Our methodology is driven by the objective of simulating somewhat realistic time series based on our lidar observations. We wanted to control the number of waves, range and distribution of*

*their random frequencies and phases. We tried to reconcile between an idealistic approach based on the superposition of a number of waves but using harmonics of a fundamental frequency which yields a set of waves with a perfectly smooth (but unrealistic) power law spectrum and the approach of fractional Brownian/Gaussian noise which is more realistic but also an idealised paradigmatic model to an extent. In order to make this simulation more realistic and improve its approximation of gravity waves time series, the frequencies are statistically independent with uniformly distributed random values, selected within the range [10min, 6h], which is typical for gravity waves. Each superposed sine wave contributes to the overall signal independently of the others (no interaction), this leads to non-intermittent, monofractal (self-similar) behaviour. Additionally, the sum of these contributions tends to exhibit asymptotically quasi-Gaussian statistics according to the principles of the central limit theorem (Billah and Shinozuka, 1990; Kirchner, 2005). Thus our simulation provides us with appealing characteristics including: random frequencies and phases (due to different gravity wave sources), reasonable number of waves based on observations (see Sica and Russel 1999), quasi-Gaussian statistics (Keisler and Rhyne 1976) and a power law spectrum that resembles that of observed and theoretically predicted spectra. In addition, having Gaussian statistics is only relevant for the Haar Structure Function method, while the other methods are not affected by the simulation statistics.*

*In short, we have presented a simple simulation of sinusoids to better test the tools and simplify the interpretation. Adding more variables also complicates the interpretation of the results. Our goal is to eventually apply this work to observations, of gravity waves, where we know that various processes, such as nonlinearity or intermittency, will affect the resulting spectra. In this case, it will be quite difficult to distinguish whether the change in slope is due to such processes or due to the bias error in the analysis method.*

Also of concern is the gap model. The statistics of the model are perhaps not as clear as they seem. The problem is that when the probability of gaps becomes large enough, many join together to become "supergaps", ultimately the left-over nongaps - where the data are sampled – may be a fractal set, see Mandelbrot's "trema" constructions (and theory) in [5]. The thing is that the fractal dimension then becomes a crucial characteristic of the sampling, and thus of the biases. Sampling data with fractal holes/gaps is indeed highly pertinent since it seems to be a fairly general problem with many geophysical data sets. However it requires its own study: how does the fractal dimension of the sampling affect the analyses?

*We agree, "supergaps" exist in observations because they are directly related to observational or experimental problems. Consider, in the case of a lidar, an aircraft passing overhead for 30 seconds or an experimental failure for two hours. Neither of these phenomena can easily be described by multifractal analysis because they are not normally distributed natural phenomena. Our approach reasonably describes both of these kinds of gaps.*

*Clouds could perhaps be described fractally, however, it is unclear how much benefit that would lend the current study. As lidar observationalists, we measure when the sky is mostly clear. As*

*a result, we tend to have random "occasional" gaps from clouds rather than a representative sample of cloud cover.*

*We cite in the manuscript a relevant study to this problem, which was done by Munteanu et al. (2016). They studied the effect of changing the size of a single large gap in the center of the time series with similar spectral characteristics to our simulation. With increasing size of this gap, the spectra of both FFT and DFT exhibit monotonically decreasing amplitudes, while the Z-Transform and Lomb-Scargle show slight increase of the average level of the spectrum. A significant change of the spectral slope was not evidently presented.*

*This is an important remark, we will consider addressing these kinds of fractal gaps and their distributions in future work.*

Haar: I was disappointed that the Haar technique was not further discussed since it is the only one that (largely) avoids most the problems discussed above (I did note that the biases reported in the paper are apparently the smallest for the Haar, as found already in [6] and reiterated in [2]). Although it was mentioned that it is easy to apply to data with gaps, the reference to this nonuniform algorithm should be given (it is in appendix of [7]). The Haar method also allow for the determination of the intermittency corrections (the entire codimension function in fact).

*We are glad to apply the HSF for the first time in our community. Indeed, we have demonstrated that it provides the least biased results compared to commonly used techniques in our field. We have also acknowledged the reference to both the nonuniform algorithm and its ability to determine the moment scaling function K(q) in the revised manuscript.*

References

1) Lovejoy, S., Scaling, dynamical regimes and stratification: How long does weather last? How big is a cloud? . Nonlinear Processes in Geophysics 2023. 30: p. 311–374.

2) Hébert, R., K. Rehfeld, and T. Laepple, Comparing estimation techniques for temporal scaling in palaeoclimate time series. Nonlin. Processes Geophys., 2021. 28: p. 311–328.

3) Springford, A., G.M. Eadie, and D.J. Thomson, Improving the Lomb–Scargle Periodogram with the Thomson Multitaper. . The Astronomical Journal,, 2020. 159 p. 205.

4) Pinel, J. and S. Lovejoy, Atmospheric waves as scaling, turbulent phenomena. Atmos. Chem. Phys. , 2014. 14: p. 3195–3210.

5) Mandelbrot, B.B., The Fractal Geometry of Nature. 1982, San Francisco: Freeman.

6) Lovejoy, S. and D. Schertzer, Haar wavelets, fluctuations and structure functions: convenient choices for geophysics. Nonlinear Proc. Geophys. , 2012. 19: p. 1-14.

7) Lovejoy, S., A voyage through scales, a missing quadrillion and why the climate is not what you expect. Climate Dyn., 2015. 44: p. 3187-3210.

*Our References:*

1) *Lovejoy, S. and Schertzer, D.: Haar wavelets, fluctuations and structure functions: convenient choices for geophysics, Nonlinear Processes in Geophysics, 19, 513–527, https://doi.org/10.5194/npg-19-513-2012, 2012.*
2) *Lovejoy, S.: A voyage through scales, a missing quadrillion and why the climate is not what you expect, Climate Dynamics, 44, 3187–3210, https://doi.org/10.1007/s00382-014-2324-0, 2014*
3) *Springford, A., Eadie, G. M., and Thomson, D. J.: Improving the Lomb–Scargle Periodogram with the Thomson Multitaper, The Astronomical Journal, 159, 205, https://doi.org/10.3847/1538-3881/ab7fa1, 2020.*
4) *Dewan, E. M. and Good, R. E.: Saturation and the "universal" spectrum for vertical profiles of horizontal scalar winds in the atmosphere, Journal of Geophysical Research, 91, 2742, https://doi.org/10.1029/jd091id02p02742, 1986.*
5) *Smith, S. M., Fritts, D. C., and VanZandt, T. E.: Evidence for a saturated spectrum of atmospheric gravity waves, Journal of the Atmospheric Sciences, 44, 1404–1410, https://doi.org/10.1175/1520-0469(1987)044<1404:EFASSO>2.0.CO;2, 1987.*
6) *Weinstock, J.: Nonlinear Theory of Gravity Waves: Momentum Deposition, Generalized Rayleigh Friction, and Diffusion, Journal of Atmospheric Sciences, 39, 1698 – 1710, https://doi.org/https://doi.org/10.1175/1520-0469(1982)039<1698:NTOGWM>2.0.CO;2, 1982.*
7) *Weinstock, J.: Saturated and unsaturated spectra of gravity waves and scale-dependent diffusion, Journal of the Atmospheric Sciences, 47, 2211–2226, https://doi.org/10.1175/1520-0469(1990)047<2211:SAUSOG>2.0.CO;2, 1990*
8) *Gardner, C. S.: Diffusive filtering theory of gravity wave spectra in the atmosphere, Journal of Geophysical Research, 99, 20 601,https://doi.org/10.1029/94jd00819, 1994.*
9) *Hines, C. O.: The Saturation of Gravity Waves in the Middle Atmosphere. Part II: Development Of Doppler-Spread Theory, Journal of the Atmospheric Sciences, 48, 1361–1379, https://doi.org/10.1175/1520-0469(1991)048<1361:TSOGWI>2.0.CO;2, 1991.*
10) *Dewan, E.: The saturated-cascade model for atmospheric gravity wave spectra, and the wavelength-period (W-P) relations, Geophysical Research Letters, 21, 817–820, https://doi.org/10.1029/94gl00702, 1994.*
11) *Pinel, J. and Lovejoy, S.: Atmospheric waves as scaling, turbulent phenomena, Atmospheric Chemistry and Physics, 14, 3195–3210, 2014.*
12) *Billah, K. Y. R. and Shinozuka, M.: Numerical method for colored-noise generation and its application to a bistable system, Phys. Rev. A,42, 7492–7495, https://doi.org/10.1103/PhysRevA.42.7492, 1990*

13) Kirchner, J. W.: Aliasing in 1/f α noise spectra: Origins, consequences, and remedies, Phys. Rev. E, 71, 066 110, https://doi.org/10.1103/PhysRevE.71.066110, 2005.

14) Sica, R. J. and Russell, A. G.: How many waves are in the gravity wave spectrum?, Geophysical Research Letters, 26, 3617–3620,https://doi.org/10.1029/1999gl003683, 1999.

15) Keisler, S. R. and Rhyne, R. H.: An assessment of prewhitening in estimating power spectra of atmospheric turbulence at long wavelengths,Tech. rep., NASA Langley Research Center, https://ntrs.nasa.gov/api/citations/19770005667/downloads/19770005667.pdf, 1976.

16) Munteanu, C., Negrea, C., Echim, M., and Mursula, K.: Effect of data gaps: comparison of different spectral analysis methods, Annales Geophysicae, 34, 437–449, https://doi.org/10.5194/angeo-34-437-2016, 2016.

---

## Referee Report (RR1)

**More comments on Assessing Atmospheric GravityWave Spectra in the Presence of Observational Gaps**

Mohamed Mossad, Irina Strelnikova, Robin Wing, and Gerd Baumgarte

Thank you for the generally useful clarifications and responses. Below, I still have a few comments that should be addressed.

Line 48: Why does $\beta>2$ necessarily lead to bias?! It will do so if there are data gaps, but why otherwise (assuming that a window was used before taking the spectrum)? Also, the existence of low frequency variability ("longer periods that the observations..") is ubiquitous in geoscience, but why is this different than the usual problem that is solved by windowing?

Line 53: I think the problem here is that the authors have set themselves the problem of identifying periodic components that stand out in the presence of a scaling "background". In this case, the main methods of astronomy and seismology are not necessarily very helpful since they mostly focus on finding the exact frequency and phase of spectral peaks, not in estimating the scaling properties of "background" spectra (although there are exceptions…). The problem of accurately estimating the "background" i.e. a signal from a wide range of frequencies is more typically a turbulent, hence atmospheric, oceanic problem.

Line 74: The L-S and MTM together are still poor when the spectrum is scaling and with gaps, see comment on line 110 below.

Line 93: As mentioned in my previous comment, determining spectra from series with missing data using interpolation will lead to a bias, unless the interpolation method has the same exponent $\beta$ as the series i.e. in general, fractal (not linear) interpolation is needed. But the right exponent is needed before such interpolation can be done! Perhaps a "bootstrap" method of iteratively interpolating with closer and closer approximation fractal would be possible…

Line 110: As indicated in the previous comments, the problem with Lomb-Scargle is that there is massive spectra leakage when either the spectral spike to too big, or the low frequencies have too much power (the exponent $\beta$ is too big, $\beta \approx >1-2$). This is because L-S is a regression technique that does not conserve the total spectral power. Adding MTM is not justified when there is missing data since the weighting functions are not orthogonal on nonuniform bases. The MTM often performs very poorly in scaling series with gaps.

Line 125: The Haar order q = 2 exponent is exactly equal to $\beta-1$ irrespective of the size of the intermittency correction: the exact general result is $\xi(2)=2H-K(2)$ whereas the result for spectral exponent is spectral exponents are exactly $\beta = \xi(2)+ 1$. The Haar fluctuations are very good for scaling processes, but are not optimal if there are large periodic components superposed on it.

Line 179: I don't understand the simulation. There are 35 frequencies and 35 phases. We are told that the frequencies are chosen from a uniform probability distribution, but what about the phases? Please write down the theoretical spectrum that this model generates. It shouldn't be too difficult, and it is needed to clarify the properties of the process which are not self-evident.
There are also potential issues of convergence since using only 36 sinusoids for each simulation, seems like a small number.
At the moment, I can't evaluate the model. The authors haven't clearly responded to this key question raised in the earlier comments.

Line 243: I don't see how aliasing can enter here. Isn't the lowering of the high frequencies and raising of low frequencies simply due to the smooth (linear) gap filling line having a paucity of high frequencies compared to the

signal?  We're replacing a signal with lots of high frequencies with one with only low frequencies.  That's why interpolation is a bad idea!

Line 335: Spectral leakage occurs for any finite nonperiodic signal, it doesn't matter what the lower frequencies would have been had they been present.

---

## Author Response (AR2)

**More comments on Assessing Atmospheric Gravity Wave Spectra in the Presence of Observational Gaps**

Mohamed Mossad, Irina Strelnikova, Robin Wing, and Gerd Baumgarten

Thank you for the generally useful clarifications and responses. Below, I still have a few comments that should be addressed.

*We thank the reviewer for their insightful comments. We repeat the reviewers' concerns and provide our respective responses in italics. The changes take place in the revised manuscript.*

**Line 48:** Why does b>2 necessarily lead to bias?! It will do so if there are data gaps, but why otherwise (assuming that a window was used before taking the spectrum)? Also, the existence of low-frequency variability ("longer periods that the observations..") is ubiquitous in geoscience, but why is this different from the usual problem that is solved by windowing?

*We did not apply a window in order to see the leakage effects for Beta> 2 (see Klis 1994, Deeter and Boynton 1982), since the effect of windowing has been studied in many other papers. Low frequency variability is an example of the same leakage problem. The prewhitening and postdarkening technique is proven to fix this leakage problem without bringing a huge loss of amplitude/power.*

**Line 53:** I think the problem here is that the authors have set themselves the problem of identifying periodic components that stand out in the presence of a scaling "background". In this case, the main methods of astronomy and seismology are not necessarily very helpful since they mostly focus on finding the exact frequency and phase of spectral peaks, not in estimating the scaling properties of "background" spectra (although there are exceptions…). The problem of accurately estimating the "background" i.e. a signal from a wide range of frequencies is more typically a turbulent, hence atmospheric, oceanic problem.

*We have replaced "such as astronomy and seismology" with "in other branches of geophysics."*

**Line 74:** The L-S and MTM together are still poor when the spectrum is scaling and with gaps, see comment on line 110 below.

*We have removed the following text to avoid confusion: "however, a combination of both the LS and MTM seems to improve on its disadvantages (Springford et al., 2020)."*

**Line 93:** As mentioned in my previous comment, determining spectra from series with missing data using interpolation will lead to a bias, unless the interpolation method has the same exponent b as the series i.e. in general, fractal (not linear) interpolation is needed. But the right

exponent is needed before such interpolation can be done! Perhaps a "bootstrap" method of iteratively interpolating with closer and closer approximation fractal would be possible…

*Presupposing the slope of the spectra by assuming a value for Beta defeats the purpose of this article in quantifying the effect of estimating the bias caused by three popular techniques for generating spectra.*
*One technique, FFT requires interpolation and linear interpolation is the simplest, straightforward, and commonly used technique. Our work quantifies the bias that one would expect using FFT with linear interpolation.*
*However, "bootstrapping" is a significant, labour-intensive addition to our current study and would be better suited to a future stand-alone manuscript.*

**Line 110:** As indicated in the previous comments, the problem with Lomb-Scargle is that there is massive spectra leakage when either the spectral spike to too big, or the low frequencies have too much power (the exponent b is too big, b≈>1-2). This is because L-S is a regression technique that does not conserve the total spectral power. Adding MTM is not justified when there is missing data since the weighting functions are not orthogonal on nonuniform bases. The MTM often performs very poorly in scaling series with gaps.

*We have addressed this point in the answer to line 74.*

**Line 125:** The Haar order q = 2 exponent is exactly equal to b-1 irrespective of the size of the intermittency correction: the exact general result is x(2)=2H-K(2) whereas the result for spectral exponent is spectral exponents are exactly b = x(2)+ 1. The Haar fluctuations are very good for scaling processes but are not optimal if there are large periodic components superposed on it.

*We rephrased line 125 to: "Hurst exponent by β = 1 + qH − K(q), since the power spectral density is a second-order moment we take q = 2."*

**Line 179:** I don't understand the simulation. There are 35 frequencies and 35 phases. We are told that the frequencies are chosen from a uniform probability distribution, but what about the phases? Please write down the theoretical spectrum that this model generates. It shouldn't be too difficult, and it is needed to clarify the properties of the process which are not self-evident. There are also potential issues of convergence since using only 36 sinusoids for each simulation, seems like a small number. At the moment, I can't evaluate the model. The authors haven't clearly responded to this key question raised in the earlier comments.

*We reiterated that "The phase shifts φi are also randomly chosen from a uniform distribution within the interval [0, 2π]." on line 185.*

*We have explained the choice of the number of waves on line 183. We are looking to replicate observations which indicate that there are a finite number of waves carrying most of the energy during a night of lidar observations. M = 35 is perfectly reasonable.*

*The theoretical spectrum $S(f)$ of our simulation is derived from the Fourier Transform of the discrete time series $x(t) = \sum_i^M f_i^{-\beta/2} \sin(2\pi f_i t + \varphi_i)$ to be $S(f) = S_0 f^{-\beta}$, where $S_0$ is an arbitrary constant, see Kirchner 2005 (Eq. 14), where in our simulation the highest frequency simulated is equal to or less than the Nyquist frequency. The spectrum will then show M peaks at the frequencies $f_i$ of the sine waves. The height of each peak in the spectrum is proportional to the amplitude squared $f_i^{-\beta}$ of the corresponding sine wave.*

**Line 243:** I don't see how aliasing can enter here. Isn't the lowering of the high frequencies and raising of low frequencies simply due to the smooth (linear) gap filling line having a paucity of high frequencies compared to the signal? We're replacing a signal with lots of high frequencies with one with only low frequencies. That's why interpolation is a bad idea!

*We have removed "(i.e., aliasing)".*

**Line 335:** Spectral leakage occurs for any finite nonperiodic signal, it doesn't matter what the lower frequencies would have been had they been present.

*We agree, we mentioned this in line 286.*

References:
- Klis, M.: Rapid variability in x-ray binaries—towards a unified description, NATO Science Series C, Springer, Dordrecht, Netherlands, 1995 edn., https://hdl.handle.net/11245/1.421015, 1994.

- *Deeter, J. E., & Boynton, P. E. (1982). Techniques for the estimation of red power spectra. I-Context and methodology. Astrophysical Journal, Part 1, vol. 261, Oct. 1, 1982, p. 337-350., 261, 337-350.*

- Kirchner, J. W.: Aliasing in 1/f α noise spectra: Origins, consequences, and remedies, Phys. Rev. E, 71, 066 110, https://doi.org/10.1103/PhysRevE.71.066110, 2005